# SIGNet: A Siamese Graph Convolutional Network for Multi-Class Urban Change Detection

Yanpeng Zhou [1,2,3], Jinjie Wang [1,2,3,*], Jianli Ding [1,2,3], Bohua Liu [1,2,3], Nan Weng [1,2,3] and Hongzhi Xiao [1,2,3]

[1] College of Geography and Remote Sensing Science, Xinjiang University, Urumqi 830046, China; zyp@stu.xju.edu.cn (Y.Z.)
[2] Xinjiang Key Laboratory of Oasis Ecology, Xinjiang University, Urumqi 830046, China
[3] Key Laboratory of Smart City and Environment Modelling of Higher Education Institute, Xinjiang University, Urumqi 830046, China
[*] Correspondence: wangjj@xju.edu.cn; Tel.:+86-158-0991-0816

**Abstract:** Detecting changes in urban areas presents many challenges, including complex features, fast-changing rates, and human-induced interference. At present, most of the research on change detection has focused on traditional binary change detection (BCD), which becomes increasingly unsuitable for the diverse urban change detection tasks as cities grow. Previous change detection networks often rely on convolutional operations, which struggle to capture global contextual information and underutilize category semantic information. In this paper, we propose SIGNet, a Siamese graph convolutional network, to solve the above problems and improve the accuracy of urban multi-class change detection (MCD) tasks. After maximizing the fusion of change differences at different scales using joint pyramidal upsampling (JPU), SIGNet uses a graph convolution-based graph reasoning (GR) method to construct static connections of urban features in space and a graph cross-attention method to couple the dynamic connections of different types of features during the change process. Experimental results show that SIGNet achieves state-of-the-art accuracy on different MCD datasets when capturing contextual relationships between different regions and semantic correlations between different categories. There are currently few pixel-level datasets in the MCD domain. We introduce a new well-labeled dataset, CNAM-CD, which is a large MCD dataset containing 2508 pairs of high-resolution images.

**Keywords:** urban change detection; siamese networks; graph convolution; category semantic information; multi-class change detection dataset

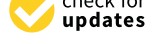



## 1. Introduction

Cities are ecosystems with a particularly close relationship to human beings and have a profound and concentrated impact on the geographical environment [1,2]. Changes in land use and land cover (LULC) as a reflection of the urbanization process and their accurate identification are essential for research in urban expansion, economic development, sociology, and related fields [3,4].

Change detection (CD), a technique for obtaining change information through multi-temporal satellite images acquired at different times, has been a hot topic of research for various scholars [5,6]. Among the traditional CD methods, multivariate alteration detection (MAD), change vector analysis (CVA), slow feature analysis (SFA), and a series of their derived methods primarily rely on the spectral information in the images to detect changes [7–9]. However, pseudo-changes caused by weather conditions, seasons, and differences in satellite sensors tend to reduce the accuracy of CD [10]. CD can also be performed based on texture, edge, and other feature information of mages, but the need for geometric registration and the tedious process limit the level of automation of CD [11]. The advancements in aerospace and computer technologies have led to continuous improvement in the quality and types of remote sensing data available for CD, and some studies have used machine learning algorithms such

as expectation maximization (EM), extreme learning machine (ELM), and random forest (RF) to automatically extract features from ground objects and perform CD [12–14]. As a branch of machine learning, the successful applications of deep learning in image classification [15,16], target detection [17,18], and semantic segmentation [19,20] provide new ideas for CD research in remote sensing images, and numerous CD network models based on deep learning have been presented [21]. Thanks to the powerful feature extraction ability of convolutional neural networks (CNN), most of the current CD network models are based on CNN [22], and some studies have also adopted deep supervision strategies, encoder-decoder architectures, and various attention mechanisms to improve the accuracy of CD [23–25]. Recurrent neural networks (RNN) are emerging in CD tasks due to their more adept handling of sequential data and reflecting the dynamic changes of images over time [26]. Given the ability of transformer networks to model long-distance contextual relationships, CD schemes based on transformer technology are becoming popular [27,28]. The Siamese network is composed of two structurally identical and weight-sharing sub-networks; it takes two samples as input and outputs a representation of the high-dimensional space and has the natural advantage of judging the difference of bi-temporal images [29].

The existing CD methods mostly focus on binary change detection (BCD), which only detects the presence or absence of changes and hardly meets the needs of diverse tasks such as urban management, environmental protection, disaster detection, LULC monitoring, etc. In contrast to BCD, multi-class change detection (MCD) involves identifying two or more categories of changes and requires detecting both the extent and type of change. MCD provides richer change information than BCD; however, the complexity of the MCD task also makes it more challenging than BCD [30]. The schematic diagrams of BCD and MCD are shown in Figure 1, where BCD represents areas of change and no change with two colors, and MCD uses different colors to represent changes in multiple categories, such as from vegetation to bare land or from water to buildings, etc.

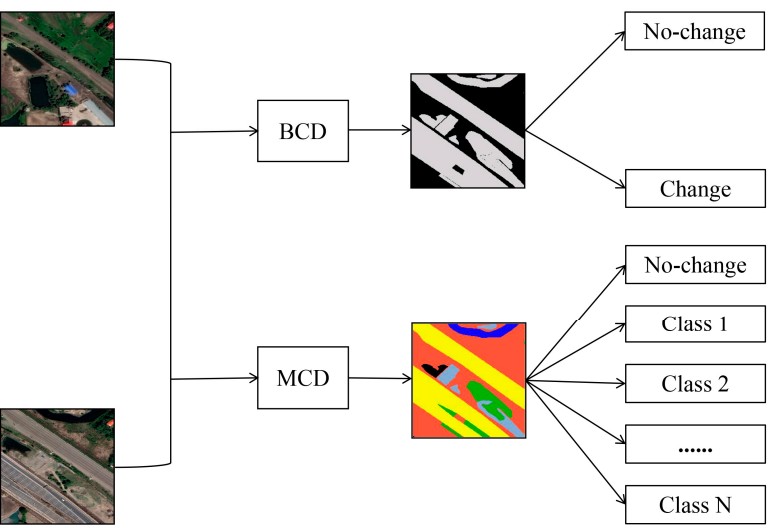

**Figure 1.** Schematic diagram of BCD and MCD.

Numerous CD methods are derived from semantic segmentation networks, and previous studies have proved that identifying contextual relationships helps in the understanding of objects [31–33]. Using correlations between the same classes (intra-class contextual information) and differences between different classes (inter-class contextual information) can strengthen the feature representation [34]. Modeling and reasoning of region relations often depend on convolutional operations, and while a single convolutional layer can capture local relations within the convolutional kernel, it falls short in capturing relations between non-adjacent regions that are far apart from each other, which requires the use of multiple convolutional layers, but this also reduces the efficiency of global

inference [35–37]. In this regard, a series of optimization methods for CNN employ full convolution, global convolution, atrous convolution, pyramid pooling, and other methods to aggregate contextual information from different regions, thus improving the ability to obtain global information [38–41]. OCRNet [42] allows contextual information to focus on the object itself, expanding the dimensionality of contextual information representations. Visual transformers (VT) segment images into non-overlapping patches and calculate the similarity between pixels using self-attention mechanisms, solving the locality problem inherent in convolutional operations [43]. However, the high computational complexity imposes limitations on the application of VT [44]. Chen et al. [35] designed a sparse token transformers structure that allows the computational complexity of the transformers to be reduced while maintaining long-range token dependency. In addition, in the process of reasoning multi-category visual relationships, there may be redundancy in the interaction between patches within the same category, and there may be a lack of semantic consistency between patches of different categories [45–47].

Graph neural networks (GNN) have attracted wide attention because of their ability to model the interrelationship between different entities and have risen to prominence in the understanding of remote sensing scenes. Yan et al. [48] used CNN to extract high-level abstract features of each vertex in the graph and GNN to propagate and aggregate the features to achieve regularized road surface extraction. He et al. [49] modeled multi-target tracking in satellite video as a graph spatiotemporal inference process to mine potential higher-order correlations between video frames. Following the proposal of the concept of graph convolutional networks (GCN) [50], several subsequent methods began to introduce graph propagation mechanisms to provide broader contextual information. Among them, GCU [51] learns graph representations from 2D features, where nodes represent regions in the image and edges represent the relationship between regions. Globe [36] performs global relational reasoning by projecting information from the feature map to the nodes in the graph. GINet [52] and SGR [53] incorporate external knowledge to assist in the reasoning of visual relations. Graph reasoning (GR) is one of the most effective techniques for establishing remote dependencies in given images. It exchanges and aggregates semantic and spatial information through weighted links between nodes, enabling region nodes with global semantic information and boundary nodes with local spatial features to characterize different entity units in the image [36]. GR has been successfully applied to computer vision tasks such as image classification [54], instance segmentation [55], and object detection [56]. Recently, some related work in remote sensing has further expanded the scope of GR applications. Su et al. [57] explored a more efficient contextual representation in semantic segmentation by introducing a dynamic graph contextual inference module. He et al. [58] parsed multiple classes of objects in remote sensing scenes as semantic entities and learned the higher-order relations through transformer-induced graph networks to achieve multimodal semantic segmentation.

Although research on MCD has appeared earlier, the datasets and algorithms for MCD have not been well developed compared to BCD [30]. Some scholars have attempted to extract multi-class change information using convolutional neural networks (CNN), but their application still has many limitations due to the lack of semantic information [59]. In recent years, various MCD datasets with different applied objectives have been made available, leading to the development of MCD algorithms and evaluation metrics. Daudt et al. [60] constructed a large-scale semantic change detection dataset (HRSCD) and designed four MCD training frameworks based on fully convolutional networks (FCNN). Yang et al. [61] introduced the SECOND dataset and further proposed an asymmetric Siamese network. Tian et al. [62] proposed a high-resolution remote sensing imagery urban MCD dataset (Hi-UCD). Several MCD networks have used encoder-decoder architectures. Ding et al. [63] proposed Bi-SRNet, which uses two additional inference blocks to infer semantic and change information, respectively. SCDNet [64] adopts multi-scale atrous convolution in the encoder and introduces attention mechanisms as well as deep supervision strategies in the decoder to achieve multi-level feature representation. Many studies decompose MCD into

two subtasks: semantic segmentation and BCD. Zheng et al. [65] proposed a deep multi-task encoder-transformer-decoder architecture (ChangeMask), in which the semantic-aware encoder is responsible for modeling contextual relationships, the transformer learns the change information, and the decoder outputs the change results. Xia et al. [66] exploited the joint features from multi-temporal images and the automatic soft fusion strategy to improve the accuracy of MCD. In the MCD dataset, the area ratios of each change category are unbalanced, and the change categories with small proportions are often difficult to accurately extract. Zhu et al. [67] proposed the Siamese global learning (Siam-GL) framework, which uses a global hierarchical mechanism and BCD masks to cope with the imbalance in the category distribution. Xiang et al. [68] designed a separable loss function to resist the effect of unbalanced labels on the MCD model. CD methods relying entirely on CNN generally struggle to capture sufficient global information from images. To address this issue, the latest research integrates global and local information through the combination of CNN and transformers. Niu et al. [69] introduced the multi-content fusion module to facilitate the extraction of change features in complex contexts by fusing foreground, background, and global information. Cui et al. [70] explored the relationship between semantic segmentation and BCD and further improved detection performance by utilizing the correlation between the two subtasks.

The lack of datasets and methods has been supplemented by recent research, but there are still some pressing issues that need to be addressed in the field of MCD. In terms of the MCD datasets, firstly, large pixel-level MCD datasets are scarce due to the difficulty of production, and most studies currently rely on the SECOND dataset and the HRSCD dataset. Secondly, existing MCD datasets often suffer from severe label imbalance, which not only increases the difficulty of the MCD task but also prevents MCD models from being compared with rapidly updating BCD models using traditional evaluation metrics, and existing CD research tends to focus on the BCD field, thereby reducing the persuasiveness of the comparison effort. Finally, many CD datasets have relatively concentrated scenes and periods, but images from different scenes and times enable adequate testing of the model's generalization performance. On the other hand, concerning the MCD approaches, the utilization of semantic information from categories by previous MCD models is inadequate, especially without fully exploiting the connections between category changes. Graph neural network has shown great superiority in expressing relationships between different entities, and using it to explore the connections between changes in categories is a novel and well-founded attempt. In response to the above problems, the main work of this paper is as follows:

(1) A Siamese graph convolutional network (SIGNet) is proposed for urban MCD tasks. SIGNet combines the outputs of the Siamese network through joint pyramid upsampling and uses graph convolution to establish reliable and robust spatial connections to achieve pixel-level MCD results.

(2) In the process of spatial relationship reasoning, we utilize the cross-attention mechanism to establish semantic associations with the category information in the dataset and incorporate the semantic association information between categories into the spatial context relationships, which provides new inspiration for MCD research.

(3) A large-scale pixel-level MCD dataset (CNAM-CD) is presented, which collects images from 12 different urban scenes over the last decade. Compared with previously released datasets, CNAM-CD has more refined labels and a more balanced distribution of categories, thus providing the possibility to evaluate each category individually.

## 2. Methods

### 2.1. SIGNet: A Siamese Graph Convolutional Neural Network

Changes in urban features are primarily caused by human activities, and the regularity of human activities leads to specific spatial and temporal connections between the changes in features. Especially in the urban remote sensing scene reflecting the real world, the remote sensing data manifold involves same-type change correlations in the spatial

dimension (e.g., roads, vegetation, and water) and different-type change correlations in the temporal dimension (e.g., water and vegetation, buildings and roads, and various policy-induced changes) [58,71–73]. Theoretically, it is feasible to capture spatiotemporal connections based on the contextual information from different change regions and the semantic information from different change categories to improve the accuracy of CD. According to the above hypothesis and findings from previous studies, we proposed SIGNet for the urban MCD task.

### 2.2. Model Architecture

The complete structure of SIGNet is presented in Figure 2. Firstly, the visual features of the image are established, and the semantic features of the category are formed, respectively (Figure 2a,b). Then, the graph projection projects visual features and category features into graph representations, respectively, and the graph re-projection projects the visual graph with category semantic information back to visual features (Figure 2c). Figure 2d illustrates the interaction process of the visual graph with the semantic graph in the graph interaction module. Finally, the change information can be obtained through a simple upsampling (Figure 2e).

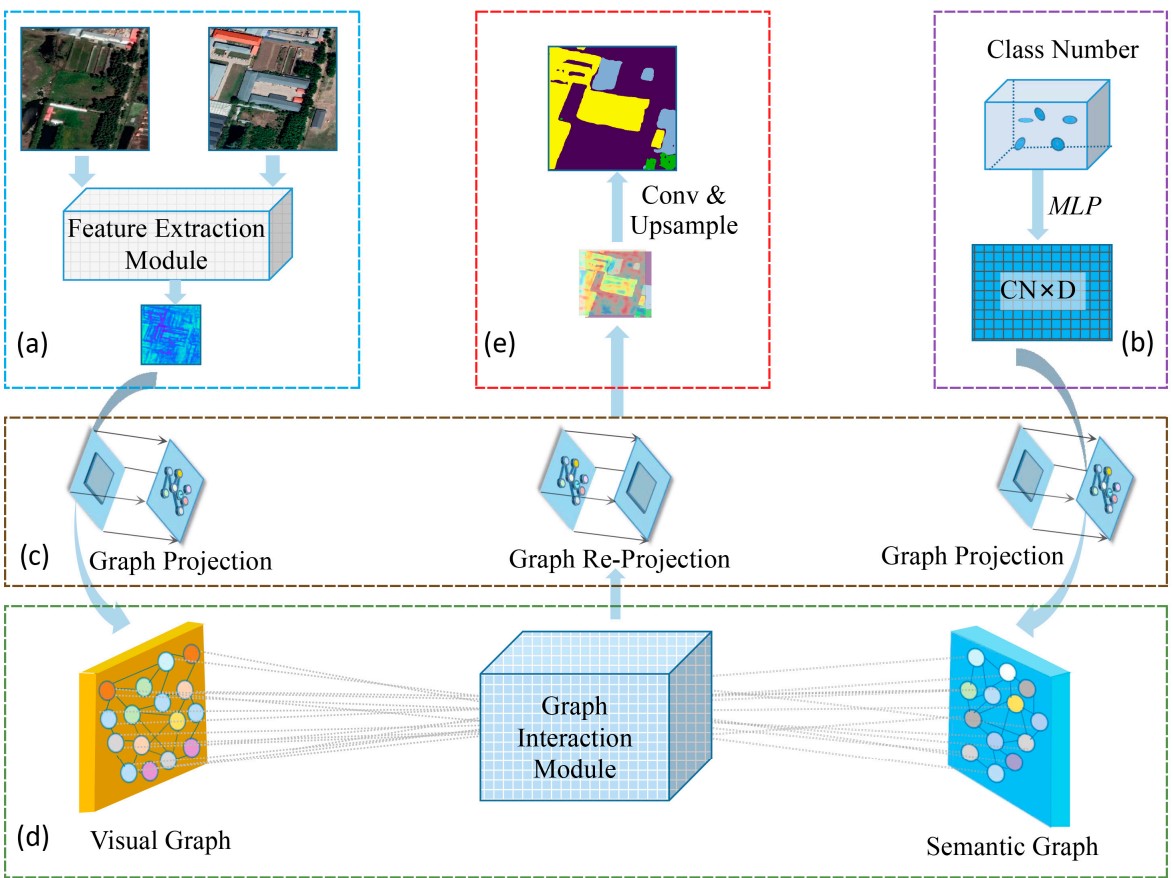

**Figure 2.** The architecture of SIGNet. (**a**) Extraction of visual features. (**b**) Formation of category features. (**c**) Global reasoning over the graph. (**d**) Interaction of visual and category graphs. (**e**) Output.

#### 2.2.1. Feature Extraction Module

This module serves to extract and merge visual features (Figure 3). At the beginning of SIGNet, the pre-trained Siamese HRNet networks are used as the backbone feature extractor. The multi-resolution parallel streaming architecture used in the HRNet preserves rich semantic information through cross-resolution interaction [74]. While HRNet can

effectively extract feature differences between two images, the Siamese network with two inputs and two subnets demands more computation and training time compared to conventional networks. To save computational resources, we use joint pyramid upsampling (JPU) for feature fusion before the subsequent operation. The Siamese backbone network can output feature maps at various scales. By using the feature maps of the first three larger scales and further performing subtraction operations between two feature maps at each scale to obtain the feature differences at different scales (Figure 3a). Then, the feature differences at different scales are upsampled to the same size, and dilated convolutions are performed using different convolution kernels (1, 2, 4, 8). Finally, after a concatenation operation, the spatiotemporal correlations are integrated into a feature map (Figure 3b) [75].

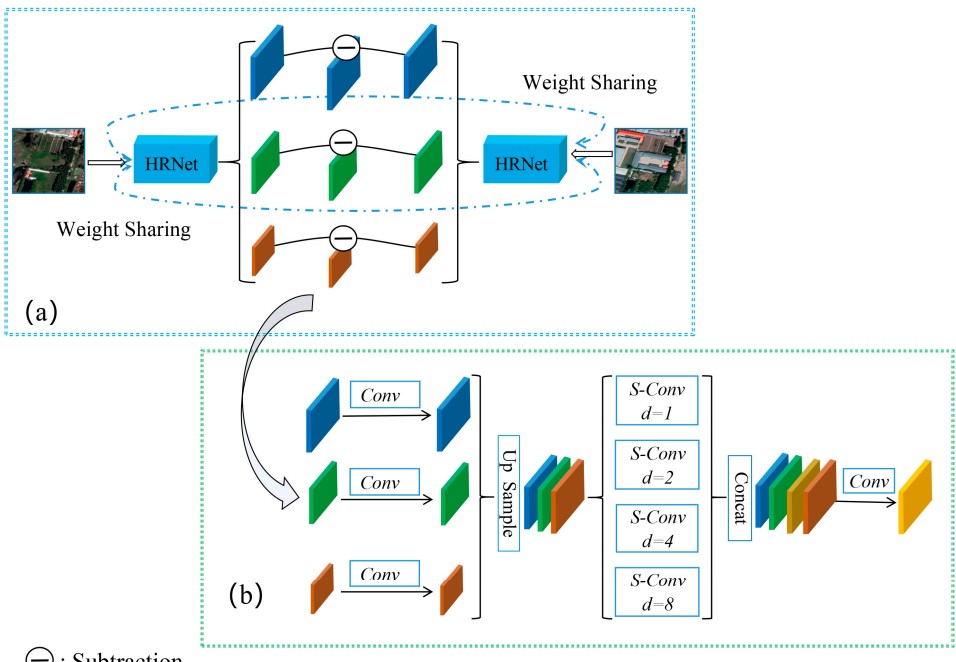

: Subtraction

**Figure 3.** The feature extraction module. (**a**) Siamese backbone network. (**b**) Joint pyramid upsampling.

In order to fully utilize the contextual and category information, we use global graph reasoning based on graph convolution to establish the contextual connections between regions (visual representation) and the semantic correlations between categories (semantic representation), as shown in Figure 4.

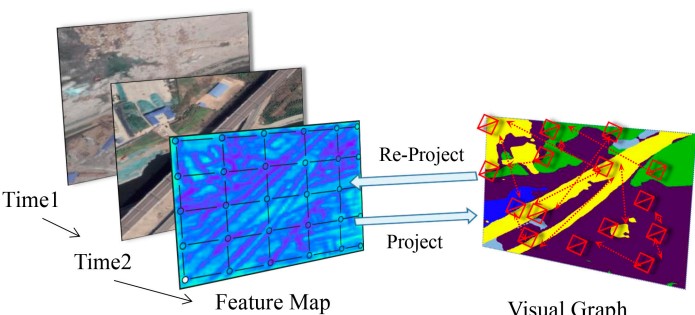

**Figure 4.** Schematic diagram of global graph reasoning.

### 2.2.2. Graph Projection

Graph projection aims to project a 2D feature map into the graph space represented by nodes and edges. Given an image feature $X \in \mathbb{R}^{H \times W \times C}$, where $H$ and $W$ correspond to the height and width of the feature map, and $C$ denotes the number of channels. Our

objective is to create a visual graph representation $V \in \mathbb{R}^{N \times D}$, where $N$ is the total number of nodes in the graph, and $D$ is the dimension of each node. Here, the image features $X$ are transformed into $V$ via $Z \in \mathbb{R}^{N \times H \times W}$, which is calculated as follows:

$$V = ZXW \tag{1}$$

where $W \in \mathbb{R}^{C \times D}$ refers to trainable parameters, and $Z$ is responsible for transforming the image features into a node in the graph.

Meanwhile, the category feature vector $L \in \mathbb{R}^{CN \times K}$ is initialized, where $CN$ denotes the class numbers, and $K$ is initialized to 300. A multi-layer perceptron can be used to transform $L$ into a category semantic feature $S \in \mathbb{R}^{CN \times D}$, where $D$ represents the dimensional number of each node:

$$S = MLP(L) \tag{2}$$

Next, the graph convolution operation is performed on the visual features of images and the semantic representation of categories, respectively [50–52]. The formula of graph convolution is as follows:

$$\widetilde{V} = \sigma(I - A_v)VW_v \tag{3}$$

where $A_v \in \mathbb{R}^{N \times N}$ is the adjacency matrix of the nodes, which is updated through gradient descent during training. $I$ is the identity matrix, which mainly serves as a shortcut connection to alleviate the difficulties of optimization. $W_v \in \mathbb{R}^{D \times D}$ are learnable parameters. $\sigma$ is a nonlinear activation function, and mapping spatial features to graphical features through $\sigma$ can obtain contextual information between nodes.

Similar graph convolution is performed on the semantic feature representation:

$$\widetilde{S} = \sigma(I - A_s)SW_s \tag{4}$$

where $A_s \in \mathbb{R}^{CN \times CN}$ is a learnable adjacency matrix and denotes semantic correlation or dependency between categories. $W_s \in \mathbb{R}^{D \times D}$ are learnable parameters. $\sigma$ is a nonlinear activation function.

A visual graph (VisG) is thus generated to encode the dependencies between different visual regions, where each node represents a visual region and the edge represents the connection between different regions. Concurrently, another semantic graph (SemG) is also created to encode the correlations among categories.

### 2.2.3. Graph Interaction Module

The graph interaction module performs interaction between the visual graph and the semantic graph. As a result of the interaction, each node on VisG obtains both contextual and category semantic information.

For a node $\widetilde{V}_i \in \mathbb{R}^D$ in VisG and a node $\widetilde{S}_j \in \mathbb{R}^D$ in SemG, the feature correlation matrix $G_{i,j}^{s2v} \in \mathbb{R}^{N \times CN}$ between the two nodes is calculated as follows:

$$G_{i,j}^{s2v} = \frac{\exp(W_v \widetilde{V}_i \cdot W_s \widetilde{S}_j)}{\sum_{cn=1}^{CN} \exp(W_v \widetilde{V}_i \cdot W_s \widetilde{S_{cn}})} \tag{5}$$

where $i \in \{1, \cdots, N\}$, $j \in \{1, \cdots, CN\}$, $W_v \in \mathbb{R}^{D/2 \times D}$ and $W_s \in \mathbb{R}^{D/2 \times D}$ are learnable matrices.

Specifically, the study here is inspired by the self-attention mechanism [42,76], as shown in Figure 5, which applies a one-dimensional convolutional layer on $\widetilde{V}_i$ and $\widetilde{S}_j$ to generate $k_i \in \mathbb{R}^D$ and $q_j \in \mathbb{R}^D$, respectively; further, the dot product operation is performed on $k_i$ and $q_j$, and the result is normalized using a softmax layer to generate association weight $A$. Another one-dimensional convolutional layer is applied to $\widetilde{S}_j$ to

generate $V_j \in \mathbb{R}^{2 \times D}$; then, $V_j$ and A are weighted and summed, and the final output is the set of values weighted by the association weights:

$$Attention(QKV) = \sum_{j=1}^{CN} soft\max(\frac{q_j k_i^T}{\sqrt{d}})v_j \tag{6}$$

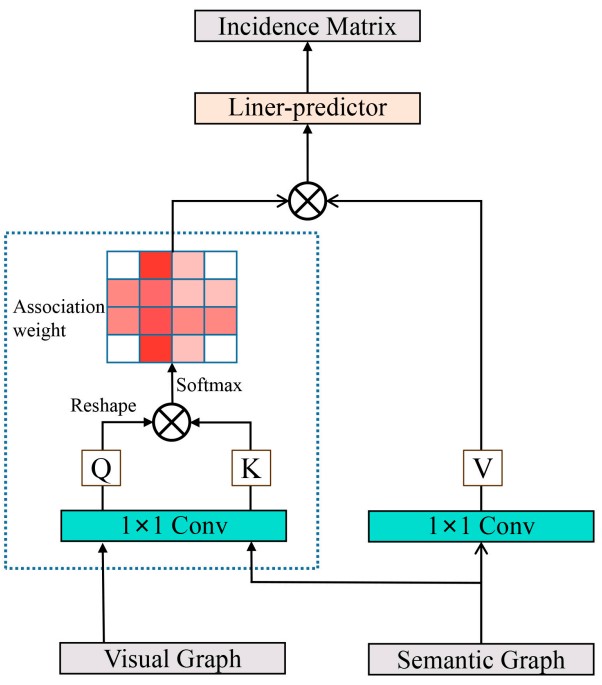

: Dot Product

**Figure 5.** Schematic of the cross-attention between category graph and visual graph.

When using $G^{s2v}$, it is possible to enhance the VisG representation with information extracted from SemG:

$$V_s = \widetilde{V} + \beta_{s2v} G^{s2v} \tag{7}$$

where $\widetilde{V}$ is the visual graph representation, and $\beta_{s2v} \in \mathbb{R}^N$ is the learnable vector.

### 2.2.4. Graph Re-Projection

Finally, the graph space representation needs to be re-projected into the coordinate space to augment the original features [51]. We reuse the projection matrix $Z$ to re-project the VisG generated by the graph interaction module into the pixel-level space to recover the image features. Given a node $V_s$ of VisG, the reverse projection can be expressed by the following equation:

$$X = Z^T V_s W_r \tag{8}$$

where $W_r \in \mathbb{R}^{D \times C}$ is a trainable weight matrix that converts the node from $V_s \in \mathbb{R}^D$ to $V_s \in \mathbb{R}^C$, and $Z^T \in \mathbb{R}^{N \times H \times W}$ denotes the transposed matrix of $Z$.

### 2.2.5. Loss Function

Cross-entropy loss is common in multi-classification tasks, and its formula is as follows:

$$loss_{ce} = -y^* + \log(\sum_{i=1}^{cn} \exp(\hat{y}_i)) \tag{9}$$

where $cn$ is the class number, $y^*$ is the label vector, and $\hat{y}_i$ is the prediction vector.

Considering that a simple summation of the losses may be insufficient, we introduce an auxiliary loss for directly supervising the learning process of the backbone network at the shallow level of the model. The final loss of the SIGNet is the sum of the two losses:

$$loss = loss_{ce} + \alpha loss_{aux} \tag{10}$$

where $\alpha$ is a learnable hyperparameter to be used to balance two losses.

### 3. Datasets and Experiment

#### 3.1. CNAM-CD: A Multi-Class Change Detection Dataset

Pixel-level MCD datasets are scarce, and we propose a new MCD dataset, CNAM-CD, to enable comparison with the BCD model using traditional evaluation metrics.

#### 3.2. Study Area

China initiated the establishment of State-level New Area (SLNA) in the early 1990s, and to date, 19 SLNAs have been established in 23 cities (Figure 6). As strategic development areas in China, SLNAs have been supported with additional policies and resources and are the most dynamic areas in cities, as well as areas with rapid and complex changes in the geographical environment [77,78]. Among the 12 SLNAs, we selected the regions with relatively balanced categories to construct CNAM-CD. The dataset contains 2503 pairs of images in GeoTiff format with a pixel size of 512 × 512, and the capture time of the images varied from 2013 to 2022 (Table 1).

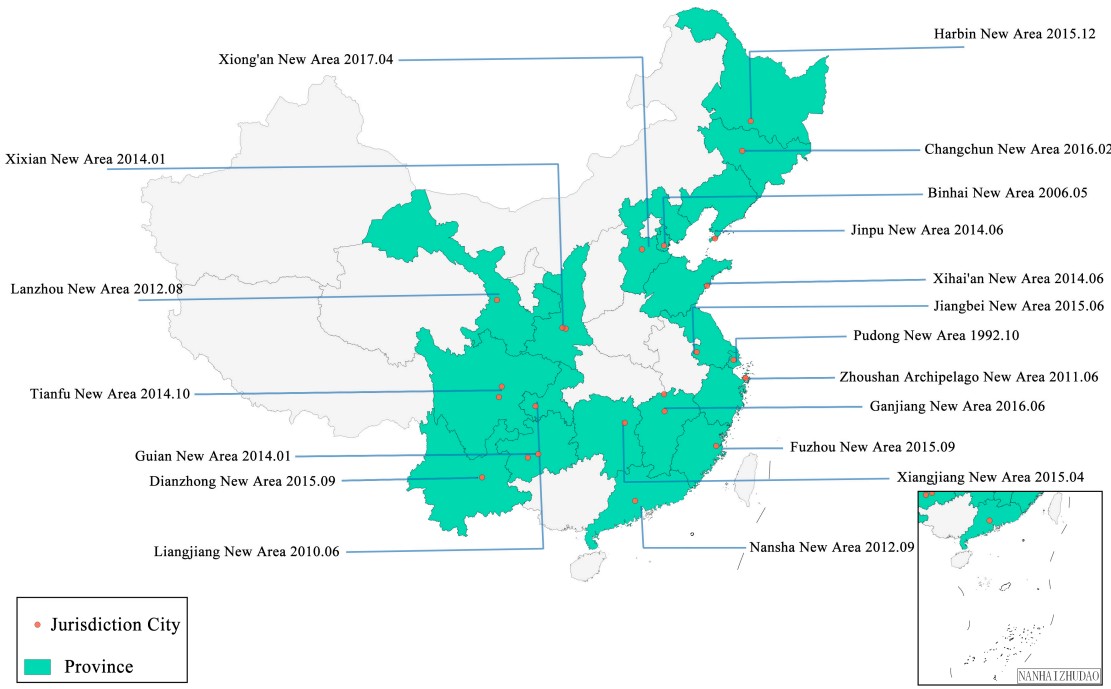

**Figure 6.** Location and establishment date of SLNAs.

**Table 1.** Details of each region and the corresponding SLNA.

| ID | Source | Jurisdiction City | Time1 (Y/M/D) | Time2 (Y/M/D) | Area (km²) |
|---|---|---|---|---|---|
| 1 | Xihai'an New Area | Qingdao | 2014/09/25 | 2019/09/18 | 14.6 |
| 2 | Jiangbei New Area | Nanjing | 2013/07/13 | 2018/10/11 | 12.7 |
| 3 | Xiangjiang New Area | Changsha | 2018/04/07 | 2021/05/09 | 12 |
| 4 | Binhai New Area | Tianjin | 2015/09/13 | 2020/06/05 | 20.5 |
| 5 | Dianzhong New Area | Kunming | 2018/03/03 | 2022/01/05 | 18.7 |
| 6 | Zhoushan Archipelago New Area | Zhoushan | 2018/03/13 | 2022/04/07 | 13 |
| 7 | Harbin New Area | Harbin | 2015/06/13 | 2021/05/19 | 18 |
| 8 | Tianfu New Area | Chengdu &Meishan | 2020/02/19 | 2021/04/29 | 16.6 |
| 9 | Xixian New Area | Xi'an &Xianyang | 2014/08/25 | 2021/11/22 | 18.4 |
| 10 | Xiong'an New Area | Baoding | 2015/08/22 | 2021/06/19 | 15.7 |
| 11 | Changchun New Area | Changchun | 2016/07/03 | 2020/06/11 | 19.9 |
| 12 | Ganjiang New Area | Nanchang &Jiujiang | 2017/12/26 | 2020/11/15 | 18.7 |

### 3.3. Data Sources and Categories

After comparing various satellite images, we chose Google Earth level 19 images. On the one hand, SLNAs cover a wide range, and the accurate identification of feature categories depends on the resolution of the images, and Google Earth's level 19 product provides very high-resolution images of 0.5 m per pixel. On the other hand, due to the large and dispersed study areas and the need to balance the categories in the label, Google Earth images are easier to acquire and process, thereby saving a lot of effort in the study area selection and image pre-processing [79]. Nevertheless, it should be additionally pointed out that users of images from Google Earth must comply with these terms and conditions as outlined on the Google Earth website (https://www.google.com/permissions/geoguidelines/, accessed on 7 March 2023). Due to differences in sensors, camera angles, weather conditions, seasons, and scenes, considerable differences occurred in the appearance (hue, saturation, contrast, brightness, etc.) of the bi-temporal images from Google Earth, making the dataset challenging and placing higher demands on the CD model. Figure 7 shows that the features in CNAM-CD are classified as bare land, vegetation, water, impervious surfaces (buildings, roads, parking lots, squares, etc.), and others (clouds, hard shadows, clutter, etc.), which have been widely used in previous urban-related research [80].

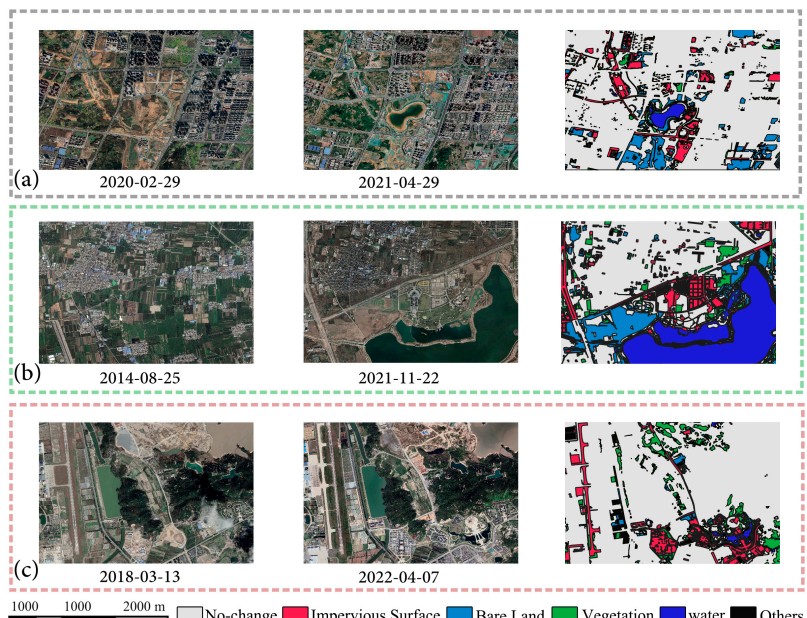

**Figure 7.** Example of regions and corresponding labels in CNAM-CD dataset. (**a**) Tianfu New Area. (**b**) Xixian New Area. (**c**) Zhoushan Archipelago New Area.

### 3.4. SECOND Dataset

The SECOND comprises 4662 pairs of high-resolution images, of which only 2968 pairs are publicly available. These images were collected from multiple platforms and sensors in Hangzhou, Chengdu, and Shanghai, and each image is 512 × 512 pixels in size. The CNAM-CD dataset focuses on growth changes, while the SECOND dataset focuses on growth and extinction changes. Specifically, the change features in SECOND are divided into six categories: non-vegetated surface, low vegetation, trees, water, buildings, and playgrounds, among which non-vegetated surface mainly represents bare land and impervious surface (Figure 8).

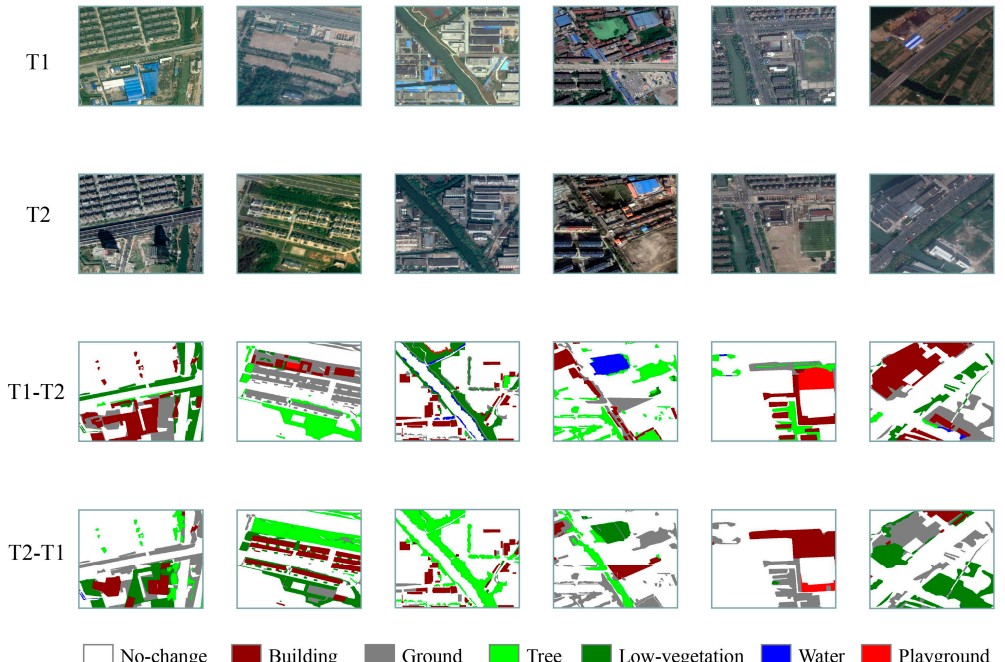

**Figure 8.** Example of images and corresponding labels in SECOND dataset.

### 3.5. Categorical Distribution

Figure 9 shows the difference in category distribution between the two datasets. Overall, the proportion of the change category in CNAM-CD is 6.5% higher than that in SECOND (Figure 9a,b). The distribution of change categories in CNAM-CD is more balanced than that of the SECOND dataset. In the SECOND dataset, N.v.g. Surface accounts for the largest proportion of the area of change at 43%, while Playground accounts for the smallest proportion at 0.38%, a difference of 113 times. In the CNAM-CD dataset, bare land has the largest proportion, accounting for 34.12% of the area of change, while water bodies have the smallest proportion, accounting for 7.83%, a difference of 4.3 times (Figure 9a,b). It should also be noted that since the existing method for identifying water bodies is relatively mature, areas with too many water bodies were discarded in the selection of data sources, and each SLNA selected contains all the categories to make the dataset more balanced (Figure 9c). The difference in category distribution between the two datasets serves as an important basis for choosing different evaluation metrics in subsequent work.

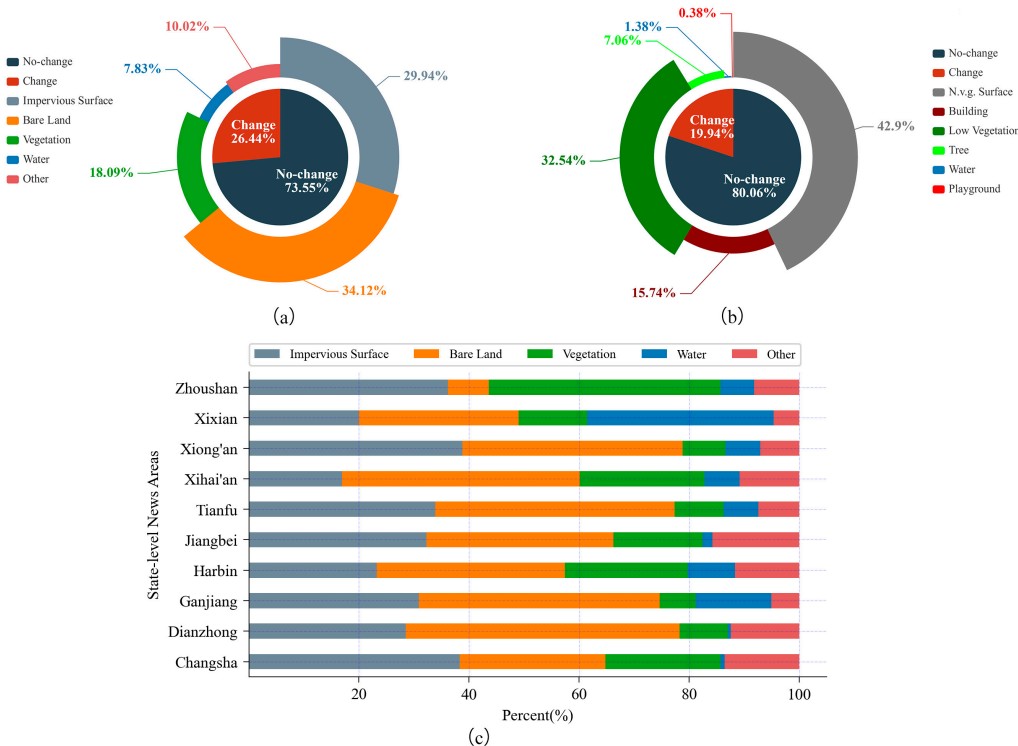

**Figure 9.** The category distribution of the two datasets. (**a**) CNAM-CD dataset. (**b**) SECOND dataset. (**c**) Distribution of categories for each SLNA in the CNAM-CD dataset.

### 3.6. Experiment

3.6.1. Evaluation Metrics

The evaluation metrics commonly used in CD tasks are inherited from semantic segmentation and include pixel accuracy (PA), precision (PR), recall (RE), and intersection over union (IoU):

$$PA = (TP + TN)/(TP + TN + FP + FN) \tag{11}$$

$$PR = (TP)/(TP + FP) \tag{12}$$

$$RE = (TP)/(TP + FN) \tag{13}$$

$$IoU = (TP)/(TP + FP + FN) \tag{14}$$

where TP denotes the count of positive examples accurately identified as positive. FN represents the count of positive examples inaccurately identified as negative. FP denotes the count of negative examples correctly identified as negative. TN represents the count of negative examples inaccurately identified as positive.

We also introduce the Kappa coefficient and F1-Score, which are defined as:

$$Kappa = (Po - Pe)/(1 - Pe) \tag{15}$$

$$F1 - Score = (PR \times RE \times 2)/(PR + RE) \tag{16}$$

where $P_o$ is the overall accuracy and indicates the probability that each pixel agrees with the label. $P_e$ indicates the probability that the classification result agrees with the label due to chance. The value of the Kappa coefficient is 1 when the result is in perfect agreement with the label.

It is worth noting that in the overall evaluation of the model, mean intersection over union (MIoU), mean precision (MP), mean recall (MR) and mean F1-score (MF) for multiple categories in the CNAM-CD dataset were adopted.

Due to the significant differences in the proportion of each category in the SECOND dataset, it is unreasonable to use traditional evaluation metrics to evaluate model performance. For example, since each pixel is equivalent in the calculation of PA, the dominant unchanged pixels (80%) will result in an unreasonably high score. Similarly, in the area of change, the N.v.g. Surface category, which accounts for 42.9% of the total pixels, clearly affected the evaluation results significantly, while the Playground category, which accounts for only 0.38% of the total pixels, can hardly affect the evaluation results.

In order to alleviate the effect of label imbalance, many MCD studies used MIoU and separated kappa coefficient (Sek) to evaluate different models:

$$\text{MIoU} = 0.5 \times (\text{IoU}_1 + \text{IoU}_2) \tag{17}$$

where $\text{IoU}_1$ and $\text{IoU}_2$ denote the IoU of the change and no-change parts, respectively.

Given a multi-class confusion matrix $Q_{ij} = (q_{ij}, 1 \leq i \leq C, 1 \leq j \leq C)$, where $C$ represents the number of categories and $q_{11}$ is set to 0 to reduce the number of true positives in the no-change category, and Sek is defined as:

$$\text{Sek} = e^{(\text{IoU}_2 - 1)} \times (\rho - \eta)/(1 - \eta) \tag{18}$$

$$\rho = \sum_{i=1}^{C} q_{ii} / \sum_{i=1}^{C} \sum_{j=1}^{C} q_{ij} \tag{19}$$

$$\eta = \sum_{i=1}^{C} (q_{j+} \times q_{+j}) / \left( \sum_{i=1}^{C} \sum_{j=1}^{C} q_{ij} \right)^2 \tag{20}$$

where $q_{j+}$ denotes the row sum of $Q$, and $q_{+j}$ denotes the column sum of $Q$.

The calculation of the comprehensive score based on MIoU and Sek is as follows:

$$\text{Score} = 0.7 \times \text{Sek} + 0.3 \times \text{MIoU} \tag{21}$$

### 3.6.2. Data Enhancement

During training, data augmentation was performed by mixing two images and their corresponding label as a new sample, randomly selecting bands in the image, random vertical and horizontal flips with 50% probability, and randomly swapping the order of the images.

### 3.6.3. Training Details

Our work is based on PaddlePaddle, which is an open-source deep learning platform that provides free GPU resources for a limited time and enables easy deployment and application of deep learning models [81]. The hardware device for training is the NVIDIA Tesla V100 GPU. The Adam optimizer is used. The initial learning rate is set to 0.0002 and reduced to 90% of the original learning rate every 1000 iterations. The epoch of training is 150 times. To save computing resources and prevent overfitting, we use early stopping to end the training in advance during training. Training ends when the model evaluation does not improve for 10 consecutive epochs, and the metric corresponding to the maximum MIoU is selected for comparison.

## 4. Results

### 4.1. Model Comparison

#### 4.1.1. CNAM-CD Dataset

Existing CD work mainly focuses on BCD, while previous MCD work tends to select MCD models for comparison. Meanwhile, most BCD models inherit from semantic seg-

mentation models, and severely unbalanced labels may also limit the effectiveness of BCD models. The CNAM-CD provides the opportunity to compare with excellent BCD models using metrics commonly applied in semantic segmentation, which enhances the objectivity of the work. Therefore, we selected recently published BCD models and slightly adapted their outputs for comparison with SIGNet.

FC-Siam-Conc and FC-Siam-Diff are fully convolutional Siamese neural networks with a skip connection structure [82].

SNUNet-CD [24] is a Siamese network with dense connections.

BIT [28] introduces the transformer as the encoder and the decoder to enhance change features.

DSAMNet [83] uses the metric module to learn change maps by comparing embedded feature vectors of bi-temporal images.

CDNet [84] constructs a deconvolutional architecture to detect changes in street scenes.

DSIFN [25] consists of two networks that perform feature extraction and difference discrimination, respectively.

EncNet-CD is an improved EncNet specifically designed for the CD task, with HRNet-W18 as its backbone network [33].

Table 2 lists the accuracy of different CD models, and the SIGNet model consistently outperforms other models in most metrics. Compared to the SIGNet18 model, the SIGNet30 model, which uses HRNet-W30 as the backbone, shows a slight improvement in accuracy. Further experiments were conducted using HRNet-W48 as the backbone, but the accuracy of the model was hardly improved. Experiments have shown that continuously expanding the output structure of the backbone has limited contribution to the model but increases the number of parameters of the model. CDNet was proposed earlier and initially for CD of street scenes, and the mere deconvolution architecture is not suitable for larger-scale remote sensing scenes. As a result, CDNet has the lowest scores on most metrics; for example, the MIoU scores for the no-change and change categories are 7.52% and 15.94% lower than our proposed SIGNet18, respectively. DSIFN uses the VGGNet to extract features and the deeply supervised network to determine the differences, but the lack of grasp for global semantic information limits its performance, with MIoU only 3.82% and 0.39% higher than CDNet for the no-change and change categories, respectively. SNUNet-CD refines the features at different semantic levels through dense connections and channel attention modules, with slightly better results than the FC-Siam series networks using skip connections. DSAMNet integrates different attention modules and adopts a deep supervision strategy, outperforming other CNN-based models. ENCNet-CD performs global contextual reasoning and highlights the category information related to the scenes, and the introduction of this model can prove the validity of a model with a similar design philosophy to SIGNet, thus showcasing the benefits of incorporating global contextual information and category information into the model design. Its MIoU in no-change and change categories is 1.58% and 5.19% lower than SIGNet18, respectively. The BIT based on the transformer architecture also performed well, scoring close to ENCNet-CD on all metrics.

**Table 2.** The general accuracy achieved by the different models in the CNAM-CD dataset.

| Model | Backbone | Type | MIoU (%) | MR (%) | MP (%) | MF (%) | PA (%) | Kappa |
|---|---|---|---|---|---|---|---|---|
| Fc-Siam-Diff | Unet | All | 59.51 | 69.69 | 74.83 | 71.91 | 85.55 | 0.70 |
| | | No-change | 84.96 | 94.90 | 89.02 | 91.87 | | |
| | | Change | 53.15 | 63.38 | 71.29 | 66.92 | | |
| Fc-Siam-Conv | Unet | All | 60.62 | 74.34 | 72.59 | 73.11 | 85.09 | 0.75 |
| | | No-change | 84.78 | 92.03 | 91.50 | 91.77 | | |
| | | Change | 54.58 | 69.92 | 67.86 | 68.45 | | |
| EncNet-CD | HRNet-W18 | All | 65.86 | 78.30 | 76.65 | 77.38 | 87.83 | 0.77 |
| | | No-change | 87.40 | 93.97 | 92.59 | 93.28 | | |
| | | Change | 60.48 | 74.38 | 72.67 | 73.41 | | |
| BIT | ResNet18 | All | 66.46 | 77.26 | 78.72 | 77.80 | 87.69 | 0.76 |
| | | No-change | 86.56 | 93.45 | 92.16 | 92.80 | | |
| | | Change | 61.44 | 75.04 | 73.53 | 74.05 | | |
| DSIFN | VGGNet16 | All | 57.15 | 70.77 | 71.73 | 70.70 | 84.67 | 0.69 |
| | | No-change | 85.28 | 93.58 | 90.57 | 92.05 | | |
| | | Change | 50.12 | 65.07 | 67.02 | 65.36 | | |
| DSAMNet | ResNet18 | All | 63.41 | 72.99 | 78.67 | 75.05 | 87.90 | 0.72 |
| | | No-change | 85.85 | 94.13 | 90.70 | 92.39 | | |
| | | Change | 57.80 | 67.70 | 75.66 | 70.71 | | |
| SNUNet-CD | Unet++ | All | 62.06 | 78.18 | 71.87 | 74.26 | 84.77 | 0.71 |
| | | No-change | 83.80 | 89.84 | 92.57 | 91.19 | | |
| | | Change | 56.63 | 75.27 | 66.69 | 70.03 | | |
| CDNet | De-Conv | All | 56.08 | 70.49 | 69.53 | 69.08 | 82.52 | 0.65 |
| | | No-change | 81.46 | 90.09 | 89.48 | 89.78 | | |
| | | Change | 49.73 | 65.59 | 64.54 | 63.91 | | |
| SIGNet18 | HRNet-W18 | All | 69.45 | 79.99 | **81.12** | 80.31 | 89.51 | 0.80 |
| | | No-change | **88.98** | **95.79** | 92.55 | 94.14 | | |
| | | Change | **65.67** | 76.04 | **78.26** | 76.85 | | |
| SIGNet30 | HRNet-W30 | All | **70.33** | **81.40** | 80.90 | **81.07** | **89.63** | **0.80** |
| | | No-change | 88.93 | 95.37 | **92.99** | **94.17** | | |
| | | Change | 64.58 | **77.90** | 77.87 | **77.79** | | |

Backbone refers to the backbone network or architecture used by the model, and the All refers to all categories in the dataset. The best results are marked in bold.

Figure 10 depicts the accuracy obtained by SIGNet and other models in different categories, where the solid lines are SIGNet18 and SIGNet30. The SIGNet family of networks achieves the highest accuracy in detecting change, no change, impervious surface, bare land, and vegetation, and the accuracy of the two models is extremely close. Moreover, the SIGNet model is more advantageous in detecting the change category than the no-change category (Figure 10a,b). The most challenging CD task is the vegetation category, where the detection accuracy of various models is relatively low compared to the other categories (Figure 10e), mainly due to the highly variable appearance of vegetation and the absence of vital spectral information in the images, which is described in the discussion. The distribution of water bodies is generally curved and long with spatial correlation, and the BIT model achieves the best results in detecting changes in water bodies (Figure 10f), which highlights the advantages of the transformer in capturing long-range contextual information.

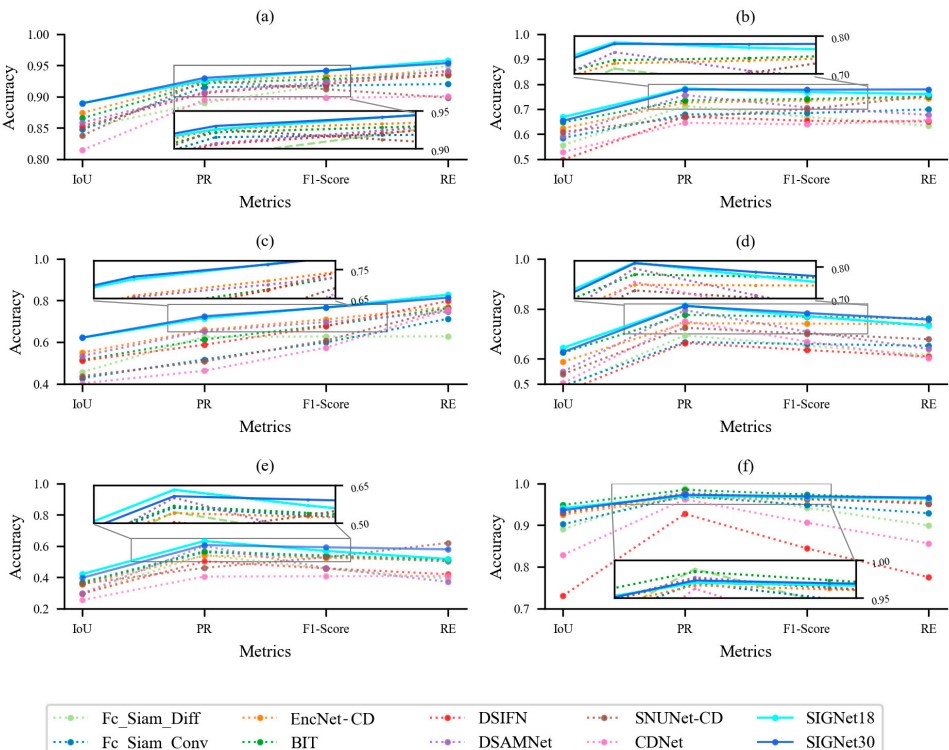

**Figure 10.** The accuracy achieved by the different models in each category. (**a**) No-change. (**b**) Change. (**c**) Impervious Surface. (**d**) Bare Land. (**e**) Vegetation. (**f**) Water.

### 4.1.2. SECOND Dataset

To validate the effectiveness of SIGNet models on MCD datasets with severe label imbalance, some BCDs and MCD models published in previous successful works were selected to conduct comparative experiments with our proposed model on the SECOND dataset.

The HRSCD series has three networks with encoder-decoder structures based on different learning strategies [60]. HRSCD-str2 treats each change category as a separate label for semantic segmentation. HRSCD-str3 trains two independent networks, where the first network performs BCD, and the second network determines the category of change. HRSCD-str4 integrates two FCNN networks into one multi-task network that performs both BCD and semantic segmentation simultaneously.

HBSCD [64] is a network with HRNet40 as the backbone.

ANS-ATL [61] explores the logical relationship among semantic categories through adaptive thresholds.

SCDNet [64] is an MCD network that uses a parallel UNet structure for multi-level feature fusion.

Table 3 shows the accuracy of different models on the SECOND dataset. Overall, except for the acceptable results of the BIT model, the performance of the BCD model was inferior to that of the MCD model. It can be seen that SIGNet18 has a higher MIoU and Score than the second-ranked SCDNet model by 1.66% and 1.63%, respectively, and Sek is 1.18% higher than the second-ranked ANS-ATL model. The HRNet backbone network performs respectably on both SECOND and CNAM-CD datasets, demonstrating its special advantage as a feature extractor for MCD tasks. In addition, among the HRSCD series networks, the one that adopts the strategy of simultaneously training semantic segmentation and BCD tasks is more advanced.

**Table 3.** The accuracy achieved by the different models in the SECOND dataset.

| Model | Backbone | MIoU (%) | Sek (%) | Score (%) |
|---|---|---|---|---|
| FC-Siam-conv | UNet | 70.10 | 12.89 | 30.05 |
| FC-Siam-diff | UNet | 70.22 | 12.51 | 29.82 |
| DSIFN | VGGNet | 69.07 | 5.90 | 24.85 |
| BIT | ResNet18 | 72.43 | 15.62 | 32.66 |
| HRSCD-str.2 [61] | FCNN | 59.70 | 5.70 | 21.90 |
| HRSCD-str.3 [61] | FCNN | 62.10 | 8.40 | 24.51 |
| HRSCD-str.4 [61] | FCNN | 67.20 | 13.00 | 29.26 |
| ANS-ATL [61] | ANS | 70.20 | 17.30 | 33.17 |
| HBSCD | HRNet-W40 | 70.40 | 15.46 | 31.94 |
| SCDNet | UNet | 72.75 | 16.86 | 33.63 |
| SIGNet18 | HRNet-W18 | 74.41 | 18.48 | 35.26 |
| SIGNet30 | HRNet-W30 | **74.64** | **18.85** | **35.59** |

Backbone refers to the backbone network or architecture used by the model. The best results are marked in bold.

### 4.2. Model Inference

We performed inference and prediction for some of the images on the CNAM-CD test set using the SIGNet model and compared the results with the BIT, SNUNet-CD, and ENCNet-CD models. Visual inspection of Figure 11 indicates that the SIGNet's inference results are closer to the labels, with lower error rates and smoother feature boundaries than other models. SNUNet-CD has a superficial understanding of the objects in remote sensing images, as evidenced by many originally smooth features being fragmented into pixel clusters. In contrast, ENCNet-CD and BIT have a deeper understanding of the objects, resulting in more natural boundaries of the land features inferred, but there are still inevitably many errors.

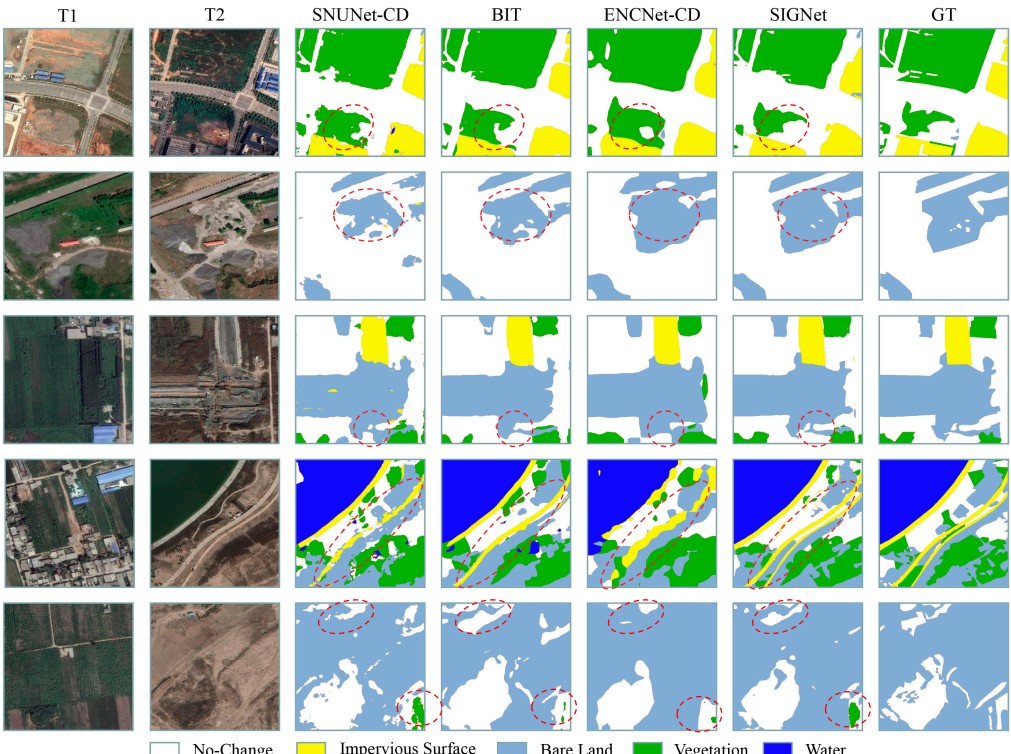

**Figure 11.** Visualization of inference results in the CNAM-CD dataset. The marked areas in the red circles reflect the differences from the ground truth.

Benefiting from a sufficient number of labels with change information, coupled with the higher robustness of the SIGNet model, as shown in Figure 12, SIGNet yields the

correct prediction for the few negative samples that are mislabeled (red circle) and for the unlabeled change features (blue circles).

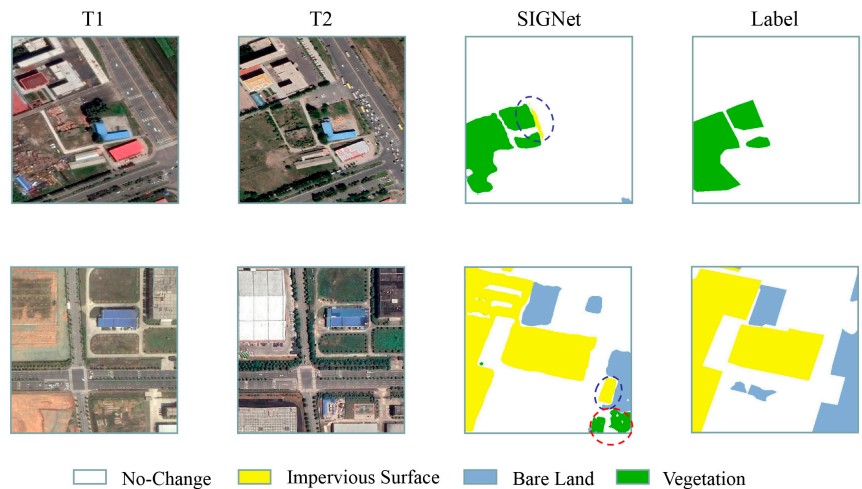

**Figure 12.** Comparison of SIGNet inference results with labels.

The SECOND dataset consists of two temporal sequences of changes, and we compared SIGNet with some models by inference. As shown in Figure 13, the first six rows represent the increasing features over time, and the last six rows represent the disappearing features over time. It can be observed that in almost every inference map, the SIGNet model produces better results than other models, as reflected in its inference results being the closest to the ground truth and having fewer erroneous pixels. Additionally, the SIGNet model demonstrated higher accuracy in recognizing small-scale categories such as playgrounds and water bodies, highlighting its advantages in identifying small-scale features. Compared with SIGNet, other models mark some pseudo-changes caused by color changes as areas of change, especially changes in playgrounds and vegetation, which are poorly identified by other models.

### 4.3. Ablation Experiment

Table 4 shows the role of the various methods used in the SIGNet model. The SIGNet is the complete model. SIGNet-GCN eliminates the graph convolution module, and SIGNet-CSI abandons the category semantic interaction module. SIGNet-AL does not use auxiliary loss. Backbone is the Siamese backbone network. Comparing the complete SIGNet model, the accuracy of the model is reduced to some extent after the removal of the GCN or CSI module, and the accuracy of the backbone network without both GCN and CSI significantly decreases.

**Table 4.** Changes in metrics after performing ablation tests for SIGNet.

| Model | GCN | CSI | AL | MIoU (%) | PA (%) | MP (%) | MF (%) | MR (%) | KAPPA |
|-------|-----|-----|-----|----------|--------|--------|--------|--------|-------|
| SIGNet | √ | √ | √ | 69.45 | 89.51 | 81.12 | 80.31 | 79.99 | 0.80 |
| SIGNet-GCN | | √ | √ | 68.30 | 89.37 | 81.00 | 79.23 | 78.53 | 0.80 |
| SIGNet-CSI | √ | | √ | 68.06 | 89.12 | 81.16 | 79.33 | 77.88 | 0.79 |
| SIGNet-AL | √ | √ | | 67.49 | 88.62 | 78.43 | 78.77 | 79.12 | 0.78 |
| Backbone | | | | 64.26 | 87.24 | 76.76 | 75.64 | 75.23 | 0.75 |

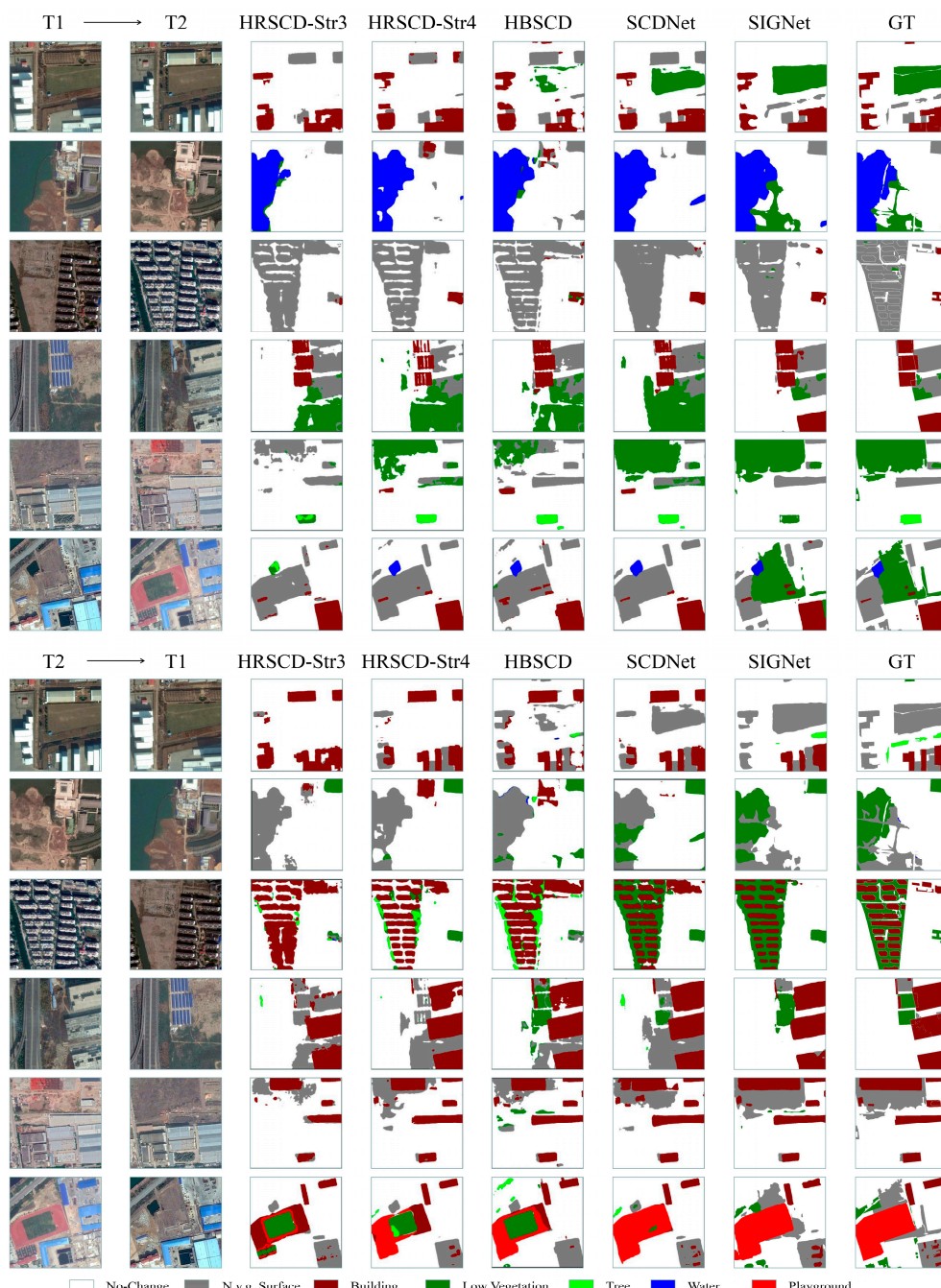

**Figure 13.** Visualization of inference results in the SECOND dataset.

## 5. Discussion

### 5.1. Attention Visualization of the Model in Different Stages

SIGNet's backbone network outputs four different scales of feature differences: scale 1, scale 2, scale 3, and scale 4, where scale 1 is $18 \times 128 \times 128$ (CHW), scale 2 is $36 \times 64 \times 64$ (CHW), scale 3 is $72 \times 32 \times 32$ (CHW), and scale 4 is $144 \times 16 \times 16$ (CHW). We explore the distribution of attention weights of the model to the images by visualizing the feature maps at different stages of SIGNet (Figure 14). In the Siamese backbone network stage, scale 1 mainly contains a large amount of texture information, and some semantic information is obtained in scales 2 and 3, but partial details are lost. Scales 1, 2, and 3 are fused by the JPU model (JPU column), which can fuse feature differences across different scales and suppress changes caused solely by the color that may mislead CD. After graph convolution and category semantic interaction (terminal column), the model concentrates

on the significant change information. The final output of SIGNet is the result learned from the auxiliary column together with the terminal column.

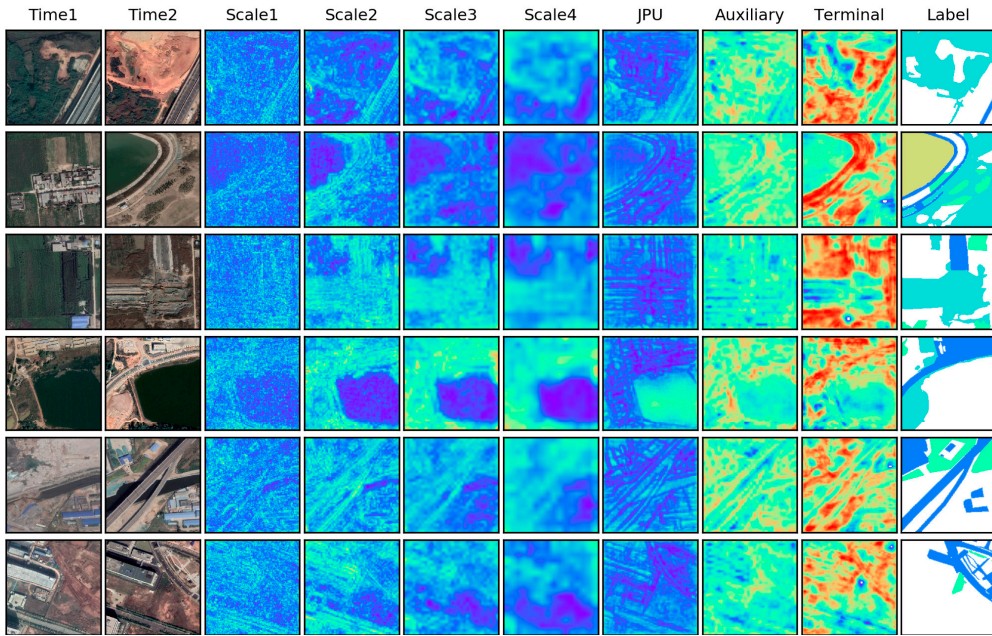

**Figure 14.** Attention maps for each stage of the SIGNet. Scale 1–4 columns are the four features output by the Siamese backbone network. The JPU column is the feature obtained after the fusion of scales 1, 2, and 3. The auxiliary column is an additional attention map used to supervise the backbone network. The terminal column is the attention map after graph convolution and category semantic interaction.

As depicted in Figure 14, in the label column, white indicates the unchanged areas, while other colors indicate the changed areas. By comparing the terminal column and the label column, it can be observed that the SIGNet model has attention self-selectivity (ASS). For example, in the first three rows, SIGNet tends to focus on changed areas, while in the last three rows, it tends to focus on unchanged areas. This adjustment of attention based on the characteristics of the objects also increases the model's recognition ability and robustness.

### 5.2. The Impact of the Characteristics of Different Categories on the Model

Apart from the proportion of categories, the characteristics of each category also impact the model's evaluation of that category. For example, affected by terrain and satellite attitude, there are geometric distortions and displacements of buildings in high-resolution satellite images (Figure 15a) and the presence of buildings under construction in rapidly urbanizing areas (Figure 15b). These factors have an impact on the CD of buildings [85]. The CNAM-CD dataset includes vegetation in different seasons, growth phases, densities, and types (Figure 15c–f). Moreover, the dataset is derived from Google Earth, which lacks infrared and other channels that are more sensitive to vegetation, making the task of detecting changes in vegetation with complex representations in the images difficult, and the accuracy for vegetation detection is lower than in other categories (Figure 10). Furthermore, to approximate real city scenes, we did not deliberately select the regions with relatively simple categories, and the CNAM-CD dataset includes other complex features such as clouds (Figure 15g), hard shadows (Figure 15h), clutter (Figure 15i,j), etc. Due to the numerous categories and the difficulty of visual interpretation, we categorize them as either change or no change and detect them as a whole.

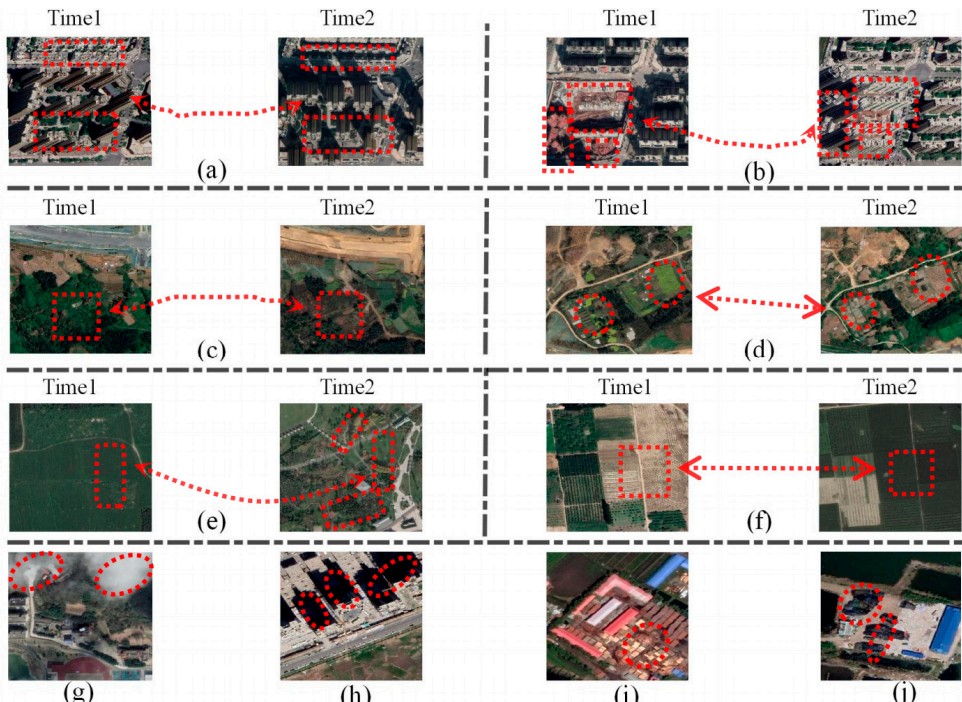

**Figure 15.** Examples of different features in CNAM-CD. (**a**) Deformation of buildings. (**b**) Construction of buildings. (**c**) Vegetation in different seasons. (**d**) Vegetation in different growth phases. (**e**) Vegetation in different types. (**f**) Vegetation in different densities. (**g**) Clouds. (**h**) Hard shadows. (**i**,**j**) Clutter.

## 6. Conclusions

This paper is the first to apply graph convolution to MCD work, and combined with the Siamese network, a model SIGNet for MCD is proposed. Through various experiments, it has been proven that the global graph reasoning method based on graph convolution can establish effective spatial associations to obtain rich contextual change information. Moreover, we use the cross-attention mechanism to map the category semantic information to the scope of change, improving the semantic consistency of bi-temporal images, and together with reasonable data augmentation methods, satisfactory CD results are achieved on MCD datasets with different category distributions. It is demonstrated that the interaction of category semantic information is helpful for the task of MCD in cities and provides new ideas for CD research in complex geographical environments. Finally, we expect that the CNAM-CD dataset presents an opportunity for researchers to pursue the development of data-hungry MCD algorithms in regions of high spatial heterogeneity. In future work, we intend to continuously update this dataset to create a larger dual-task-oriented MCD benchmark dataset that includes more categories and all of China's new state-level areas. The first version of the CNAM-CD dataset can be found at https://github.com/Silvestezhou/CNAM-CD, accessed on 3 May 2023.

**Author Contributions:** Conceptualization, Y.Z.; methodology, Y.Z.; formal analysis, Y.Z.; writing—original draft, Y.Z.; visualization, Y.Z.; writing—review and editing, J.W. and B.L.; funding acquisition, J.W. and J.D.; project administration, J.W.; data curation, N.W. and H.X. All authors have read and agreed to the published version of the manuscript.

**Funding:** This research was funded by Xinjiang Uyghur Autonomous Region Key Laboratory Open Subjects under grant 2020D04038, the National Natural Science Foundation of China Joint Fund Key Projects under grant U2003202, the Xinjiang University Doctoral Initiation Fund under grant BS190201, and Xinjiang Uygur Autonomous Region Education Department University Research Program under grant XJEDU2021Y009.

**Data Availability Statement:** The data presented in this study are available in the Conclusions of this article.

**Conflicts of Interest:** The authors declare no conflict of interest.

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
