# Peer review of "SIGNet: A Siamese Graph Convolutional Network for Multi-Class Urban Change Detection"

_remotesensing, doi:10.3390/rs15092464_

Round 1

Reviewer 1 Report

This paper proposes SIGNet, a siamese graph convolutional network, to solve the above problems and improve the accuracy of urban multiclass change detection (MCD) tasks. However, there are still some points needed to be specified clearly and solved before acceptance.

1. Figure 1 in the manuscript, the poundage of the line connecting the double time phase should be different from the other lines in Figure 1, please check that, as well as suggesting to check the drawing details of the whole figure of Figure 1, such as class1, class2 and etc, the ends of the arrows do not converge precisely in one place.

2. The Introduction section should be more abundant and the literature review should be improved. It is suggested to add some latest reasearches related to MCD,  https://doi.org/10.1016/j.isprsjprs.2021.12.005,https://doi.org/10.48550/arXiv.2011.03247.

3. In the section leading to the main work of the manuscript, the contribution of the method proposed by the authors at the methodological level in the field of MCD is emphasized, but the explanation of why the new home-made dataset method is proposed is very shallow, and it is suggested to think again about the deeper meaning of proposing this dataset.

4. Figure 2 (d) The style of the lower border is different from the upper left and right, please check and modify.

5. In the dataset section, it is mentioned that the category distribution of the SECOND dataset is uneven, and the category distribution of the dataset proposed by the authors is more balanced, so could it be because the authors have divided many features with potentially inconsistent categories into others so that they can get a more balanced category distribution?

6. The accuracy of the experimental results is evaluated in the Table 2, and it is recommended to mark the best results on which network the evaluation index is respectively.

7. Experiments were conducted on both datasets of the manuscript, and both mentioned to compare with the more effective BCD models, so why not choose  consistent networks of comparison experiments on both datasets?

Author Response

Please see the attachment and thank you to the reviewers and editors.

Reviewer 2 Report

The paper proposes a siamese graph convolutional network for multiclass urban change detection, and introduces a new dataset named CNAM-CD. The experimental results demonstrate that the proposed method achieves good performance on the CNAM-CD dataset as well as other open-resource datasets. While the authors have provided a detailed illustration of their method, there are some concerns that need to be addressed.

1.     The authors' paper outlines the strengths of using a graph convolutional network for multiclass urban change detection. It would be beneficial for the authors to elaborate on the reasons and logic behind their approach in more detail.

2.     While the authors make a valuable contribution to the change detection field by publishing their dataset, the proposed method lacks innovation. The authors should introduce some specific design choices that address the challenges of the task. Furthermore, the abstract should highlight the innovative aspects of the proposed method more explicitly.

3.     The authors have indicated that they use a graph convolutional network to capture long-range contextual information. However, it is worth discussing the differences between this approach and the transformer, which is considered a better choice for such tasks. The authors should review some transformer-based methods in the literature and compare their strengths and weaknesses to the proposed approach.

Some relevant references that the authors might consider include:

Chen K, Zou Z, Shi Z. Building extraction from remote sensing images with sparse token transformers[J]. Remote Sensing, 2021, 13(21): 4441.

Zheng Z, Zhong Y, Tian S, et al. ChangeMask: Deep multi-task encoder-transformer-decoder architecture for semantic change detection[J]. ISPRS Journal of Photogrammetry and Remote Sensing, 2022, 183: 228-239.

4.     The introduction section needs improvement in terms of the review of previous methods. The authors should provide a more thorough and objective review of the state-of-the-art methods that are closely related to their research topic. Moreover, when comparing their method with other methods, the authors should clarify the advantages and disadvantages of each method.

5.     The paper should be thoroughly proofread to correct any grammatical errors, such as "Experimental results show indicate."

Author Response

(The authors gave the same response as above.)

Reviewer 3 Report

The authors present an interesting approach to CD in urban areas. The method is well presented and sufficient experiments were undergone to prove its qualities.

One of the results of the research is the CNAM-CD dataset. If so, please provide a link to it so that it can be actually considered a contribution.

You have a display error in Table 3 (a word is hidden behind the table).

Author Response

(The authors gave the same response as above.)

Round 2

Reviewer 2 Report

It's all right! There is no extra comments of the revision.